# Nanoparticles and Nanomaterials-Based Recent Approaches in Upgraded Targeting and Management of Cancer: A Review

**DOI:** 10.3390/cancers15010162

**Published:** 2022-12-27

**Authors:** Anupama Ojha, Sonali Jaiswal, Priyanka Bharti, Sarad Kumar Mishra

**Affiliations:** 1Department of Allied Health Science, Mahayogi Gorakhnath University, Gorakhpur 273007, India; 2Department of Biotechnology, DDU Gorakhpur University, Gorakhpur 273009, India

**Keywords:** tumor heterogenicity, chemotherapeutics, microenvironment, immunomodulation, biocompatibility

## Abstract

**Simple Summary:**

According to the International Agency for Research on Cancer (IARC), in 2018, the global burden of cancer is expected to grow to 27.5 million new cancer cases by 2040. The present major area of cancer therapies includes surgery, radiation, and chemotherapy that damages normal tissues or incompletely eradicates cancer. Nanomaterials monitor in surgical resection of tumors, target chemotherapies directly and selectively to cancerous cells and neoplasms, and augment the therapeutic efficacy of radiotherapy. It decreases the risk and increases the survival of cancer patients. In the present review, our aim is to combine the different types of nanomaterial applied in different types of therapy. Researchers can be equipped with knowledge of types of nanoparticles, their mechanism of action, and how they add to existing therapies for cancer, for better cancer management.

**Abstract:**

Along with the extensive improvement in tumor biology research and different therapeutic developments, cancer remains a dominant and deadly disease. Tumor heterogeneity, systemic toxicities, and drug resistance are major hurdles in cancer therapy. Chemotherapy, radiotherapy, phototherapy, and surgical therapy are some prominent areas of cancer treatment. During chemotherapy for cancer, chemotherapeutic agents are distributed all over the body and also damage normal cells. With advancements in nanotechnology, nanoparticles utilized in all major areas of cancer therapy offer the probability to advance drug solubility, and stability, extend drug half-lives in plasma, reduce off-target effects, and quintessence drugs at a target site. The present review compiles the use of different types of nanoparticles in frequently and recently applied therapeutics of cancer therapy. A recent area of cancer treatment includes cancer stem cell therapy, DNA/RNA-based immunomodulation therapy, alteration of the microenvironment, and cell membrane-mediated biomimetic approach. Biocompatibility and bioaccumulation of nanoparticles is the major impediment in nano-based therapy. More research is required to develop the next generation of nanotherapeutics with the incorporation of new molecular entities, such as kinase inhibitors, siRNA, mRNA, and gene editing. We assume that nanotherapeutics will dramatically improve patient survival, move the model of cancer treatment, and develop certainty in the foreseeable future.

## 1. Introduction

Despite numerous current developments in tumor biology and chemotherapy, cancer still endures as a widespread and deadly disease responsible for 10 million deaths in 2020 worldwide. Tumor heterogeneity, resistance to drugs, and systemic toxicities pose major hurdles in cancer therapy. Chemotherapy, phototherapy, radiation therapy, and surgery are the major therapeutic practices used to treat cancerous cells. Along with cancerous cells, these practices also cause damage to normal cells. The principal objective of scientists and physicians is to increase the safety and efficacy of cancer therapy. Drug targeting is the best approach for cancer treatments, which contains a harmonization of acts between the devastation of cancerous and healthy tissues, with impairment to the immune system and extremely replicating cells. Nowadays scientists are working to solve these side effects by using a new branch of biotechnology, i.e., nanotechnology. Newly established nano-technological approaches, such as vigorous targeting transporter, discriminatory targeting, and delivery of drugs in tumor tissues have increased the pharmacokinetics and diminished the systemic toxicities of chemotherapies. It selectively and directly helps to target chemotherapeutics to cancerous cells and neoplasms. Expansion of drug transfer in cancer, accelerated the growth of novel nanomaterials and nanocarriers to defend the drug from fast deprivation and permit it to reach the tumor site at the correct therapeutic concentrations, without delivery to normal sites to reduce adverse effects. Along with targeted delivery nanomaterials and nanocarriers, they have also improved the therapeutic efficacy of radiation treatment and in guidance of surgical removal of tumors. Hereby it reduces mortality and increases the probability of survival of cancer patients. Figure 1 explains the types of nanoparticles in different types of cancer therapy.

The elementary configuration of nanoparticles is moderately complex, encompassing the core, surface, and shell layer typically termed as the nanoparticles (NPs) themselves. The submicron size, high surface volume ratio, enhanced targeting, and dissimilarity of NPs have made it suitable in multidisciplinary fields [1]. Nanotherapy is a technique of using engineered nanoparticles produced by nanotechnology for their targeted delivery at the disease site to reduce damage to healthy cells. Nanoparticles ranging from 10–500 nm in size, are suitable for both active and passive targeting therapies. The energy absorption and re-radiation properties of nanomaterials can be utilized for disrupting disease tissues. Solid tumors have a hypervascular system with new vessels having abnormal architecture and higher permeability. This property of solid tumors is called the enhanced permeability and retention effect (EPR effect). Nanocarriers are expressed to transport drugs either by passive targeting, taking benefits of permeability of tumor vasculature (EPR effect), or by active targeting using ligands that intensify the tumoral uptake potentially. It results in greater antitumor effectiveness, completing a net enhancement in the therapeutic index [2]. Nanoparticles are employed in therapy through the involvement of the EPR effect [3]. EPR permits the entry of nanosized antitumor agents and the magnetic properties of nanoparticles initiate an antitumor response by induction of hyperthermia under a magnetic force that begins to heat up and destroy the cancerous tissue [4,5]. Table 1 shows different classes of nanoparticles, their characteristics, mechanism, and application in cancer management.

## 2. Application of Nanoparticles in Frequently Applied Cancer Therapies

### 2.1. Chemonanotherapy

Hodgkin’s and non-Hodgkin’s lymphoma, germ cell cancer, ovarian cancer, small cell lung cancer, acute myelogenous leukemia, and choriocarcinoma are effectively treated by chemotherapy [46]. The non-specificity of chemicals used in chemotherapy causes cytotoxicity and induces multidrug resistance (MDR) in tumor cells. Poor solubility, lack of specificity, short half-life, and stem-like growth of cells are some other problems associated with chemotherapy includes. To overwhelmed these drawbacks, nanomaterial-centered chemotherapy developed utilizes the large surface area of nanomaterial for manufacturing nano-drugs that can sustain bioavailability, better specificity, larger loading capacity, less cytotoxicity to normal tissue, unique drug release patterns, and longer half-life period [47,48]. Mixed micelles, liposomes, nanostructured lipid carriers, nano lipid-drug conjugates, solid lipid nanoparticles, and nanoemulsions are lipid-based nanoparticles shown some encouraging results for use in oral anticancer drug delivery through a nanotechnological approach [49].

Internal hydrogen peroxide (H_2_O_2_) of tumor cells is transformed into toxic hydroxyl radicals (OH) by chemodynamic therapeutic substances responsible for the killing of those cells. Chemodynamic therapy (CDT) is a Fenton and Fenton-like reaction-based new in situ strategy to enhance anticancer efficacy through the generation of free radicals or reactive oxygen species and oxygen such as OH or O_2_ [50]. Through nanotechnology, the Fenton-based nanoparticles are developed which are primarily iron or metal-based (Mn^2+^, Cu^2+^, and Ti^3+^ ions) or organic NPs [51,52,53]. The large surface area, tumor-targeting ability, and high reactivity of nanoparticles are powerful apparatuses for the production of free radicals. These strategies are used to destroy cancer cells directly or improve therapeutic proficiency by combination therapy, or in monitoring/diagnose of tumor cells. Therapeutic efficiency is improved by GSH depletion or H_2_O_2_ concentration elevation in tumor cells.

### 2.2. Photonanotherapy

Phototherapy is a non-toxic, and vulnerable technique to abolish cancer cells when light waves of a certain wavelength are used with suitable triggering agents. It is mostly restricted by its quality to diseases that are deeply located in the skin. It helps in the diagnosis and treatment of numerous kinds of cancers. Some free radicals released by chemotherapy induce the death of tumor cells and release the hypoxic illness in the tumor micro-environment that is posing main obstacles in photodynamics and radiotherapy. The mixture of nanomedical agents and phototherapy shows better results than separate treatments. We are focusing in detail on different categories of phototherapy techniques to treat cancer.

#### 2.2.1. Photothermal Therapy (PTT)

PTT utilizes the nearby-infrared laser to produce warmness for the current abscission of cancer cells after laser irradiation on primary tumor or locally metastasized cancer cells [54]. It has nominal insensitivity, great specificity, and specific spatial-sequential selectivity making it a suitable choice for cancer treatment. In numerous preclinical animal experiments, the use of nanoparticles in photothermal therapy encourages therapeutic effectiveness in metastatic cancer [55]. It could be used either alone or along with imaging, radiotherapy, immunotherapy, and chemotherapy [56]. In the current state, no possible explanation of the molecular machinery of constraining cancer metastasis by PTT is cited. The photothermal nanomaterial (PTN) used along with PTT including inorganic noble metals, graphene, or carbon nanotube shows long-term toxicity after clinical implications. To overwhelm these margins, extra refined strategy and production with accurate targeting, influential imaging, and synergistic properties, to increase the beneficial consequence of cancer metastasis is required. The heat distribution inside the cancerous cell is not homogeneous, reducing the efficacy of PTT to completely remove all the cancerous cells [57]. To increase the preventive ability of PTN, it can be designed with precise directing and profound diffusion competency in tumor tissues.

Highly efficient conversion is difficult to achieve for photothermal agents. In a research study, daptomycin (Dap) micelles-stabilized palladium nanoflowers (Dap-PdNFs; 106 nm) were created and irradiated showing high photothermal conversion efficiency. HeLa cells and HT-29 cells of Dap-PdNFs reached 95% cell viability, in the absence of near-infrared (NIR) light, showing nanoflower’s biocompatibility. The rate of inhibition on HeLa cells was 71% under 808 nm near-infrared irradiation [58]. Another study done by Zhao et al. [59], prepared bovine serum albumin (BSA) coated Ag2S nanoparticles as a photothermal agent. The photothermal conversion was highly efficient with NIR absorption of approximately 18.89%. The Ag2S nanoparticles induced effective apoptosis of tumor cells under near-infrared light (808 nm). This can be an effective PTT with the successful targeting of tumor cells. In another synergistic study, a combination of photothermal and immunotherapy was developed with the help of imaging guidance and MRI technique to treat metastatic pancreatic cancer. The tumor microenvironment nanoplatform was designed with the coating of DSPE-PEG and indocyanine green (ICG) onto imiquimod (IMQ) loaded amorphous iron oxide nanoparticles (IONs). IMQ@IONs/ICG act as delivery vehicles for ICG AND IMQ and catalyze tumor microenvironment modulation. The interventional photothermal therapy treatment in H7 pancreatic tumor mice models indicates the efflux of IMQ, which triggered antitumor immunity. This decreased metastases and increased the CD8+ population [60].

#### 2.2.2. Photodynamic Therapy (PDT)

Photosensitizers excited by light produce cytotoxic radicals by contact with biomolecules (Type I pathway), or, in the Type II pathway, translate ground-state oxygen (3O2) into cytotoxic singlet oxygen (1O2). PDT utilizes visible light, a photosensitizer, and molecular oxygen to destroy cancer cells [61]. Photodynamic therapy causes the death of cancer cells by the production of reactive oxygen species. PDT can not only destroy cancer cells directly but also originate vascular impairment that hampers tumor oxygen supply. Photosensitizers are mostly metallic compounds having low toxicity, high targeting, and minimal side effects. Photofrin a 1st generation of the most commonly used photosensitizer, poses some drawbacks, such as hydrophobicity, weak absorption, low chemical purity, cutaneous phototoxicity, and in the therapeutic window, limiting its use in clinical treatment [62]. It shows non-invasive and selective cytotoxicity to malignant cells and damages them through autophagy, and necrosis, regulated by oxygen concentration, pH value, and the structure of the photosensitizer [63]. It has improved the quality of life and prolonged survival of cancer patients and is famous as a more attractive form of treatment.

Nanoparticles provided many clues to overcome the hindrances in photodynamic therapies. In recent research, mTHPC &BU@VES-CSO/TPGS-RGD nanoparticles (T-B@NP) of size 148.3 ± 2.5 nm were prepared and examined for their effects on mice models. It was found that these nanoparticles accumulated passively in tumor cells. Upon radiation exposure, mTHPC induces cell death and apoptosis. Simultaneous efflux of BU inhibited HIF-1α and declined VEGF-mediated angiogenesis that induces an effective PDT against colorectal cancer. The antitumor effect (84.2%) and an increase in the survival of mice were observed [64]. Another study done by Guan et al. [65], developed nanoparticle (NPs-Lip@PTX/CyA/Ce6) fabricated with BSA-based nanoparticles that consist of paclitaxel (PTX) and P-gp inhibitor cyclosporin A (CyA) and Chlorin e6 (Ce6) loaded Tf-modified liposomal bilayer as a shell. Upon exposure to irradiation, NPs-Lip@PTX/CyA/Ce6 demonstrated intracellular ROS formation and effective in vitro and in vivo anti-cancer therapy. Combining photothermal and photodynamic therapy with the use of a single exciting light source is a difficult task. Interesting research carried out by Cheng et al. [66], demonstrated that integration of Cu1.96S-Gd@FA nanoparticles, MR/IR dual-modal imaging, and photothermal/photodynamic therapy on one nanoplatforms could be effective in breast cancer therapy by targeting folate receptor overexpressed on breast cancer cells.

Further research should be stimulated to formulate an active amalgamation of stimulating mediators and phototherapy for the finest result, to reduce opposing things, and to stop damage to healthy tissues.

### 2.3. Radionanotherapy (RT)

In lateral chemotherapy and surgery radiation therapy are the chief helpful approaches for cancer therapy. High-intensity ionizing radiations cause hydrolysis in the target tissues resulting in the generation of different free radicals, such as superoxides O_2_, hydrogen radical H, and hydroxyl radical OH, responsible for damaging different molecules, such as DNA, proteins, and lipid bilayer resulting in the death of tumor cells. The probability of harm to the normal adjacent tissue and the development of resistance towards radiation in deeply located cancerous cells are major disadvantages of radiation therapy [67]. Along with that, the efficiency of radiotherapy is typically restricted by tumor hypoxia-related radiation resistance and is also not effective in the control of tumor metastasis, responsible for major cancer death [68].

Radionanotherapy is an evolving tool aimed to increase the effectiveness of radiotherapy by employing the usage of dendrimers in nanovectorization [69]. The use of nanoparticles in radiation therapy to act as a therapeutic or a carrier for therapeutics has occupied a major part in the improvement of radiation therapy. The targeting of cancer cells, the biodistribution of nanoparticles into the body, their ultimate toxicity, and their industrial manufacturing are major challenges associated with the use of nanoparticles [70]. The use of nanoparticles in radiotherapy has been proven to activate strong cancer immunotherapy [68].

Gadolinium-based nanoprobes are used as theranostic to improve radiotherapy efficacy [71]. The nanoparticles help in the enhancement of radiosensitization of tumor tissue, reversal of radiation resistance in tumor tissue, and enhancement of the radioresistance of the healthy tissue [72]. The exclusive physicochemical characteristics and high X-ray absorption make gold nanoparticles ideal radiosensitizers for radiotherapy [73]. High biocompatibility, slower systematic clearance, immune evasion capability, better permeation into tumor tissues, size adjustment according to requirement, and easy performance of pharmaceutical study are the major characteristics of gold nanoparticle use in radiation therapy [37]. Other nanosensitizers used are titanium oxide, silver nanoparticles, etc. NBTXR3 was the first nanoparticle composed of hafnium (Z = 79), used to treat sarcoma in humans. AGuIX, a gadolinium-based nanoparticle (Z = 64), is used in the management of brain metastases. The continued clinical trials support the feasibility, safety, and efficacy of these nanoparticles in radionanotherapy.

### 2.4. Nanosurgical Therapy

Surgery is the utmost operative conduct in the early stages of gastrointestinal and lung cancers. Blood loss, adverse reactions to medicines, and damage to organs in the body are a few complications associated with surgery. Pain, blood clot, discomfort, or other illness was reported post-surgery. Through nanotechnology, tiny biosensors could be constructed which could minimize invasion, lessen and allow faster recovery. It also saves hospitals money, reduces infection rates within the hospital, sinks the waiting lists for operations, and permits doctors to treat other patients in the same period. One of the highest attainments of nanotechnology in surgery will be an “ideal graft”, i.e., biocompatible and long-lasting “repairs” of body parts, such as arteries, joints, or even organs [74]. Nanobots are robots controlled by computers produced through nanotechnology that would be reduced for admission into the body, through openings that can not perform with the scalp and destroy individual cells without disturbing others [75]. The needle and scalpel are the surgical instruments typically affected by nanotechnology. Nanocoated surgical blades reduce trauma with a cutting-edge diameter in the region of 5 nm–1 μm. Roszek et al. [76] have explained the use of nanomaterial-coated diamond scalpels, ophthalmic surgical blades, and trephines in eye or neurosurgery have higher stability. Nanoneedles are used in ophthalmic and plastic surgery having exceptional strength, good ductility, and corrosion resistance [77]. When attached to a microscope, nanoneedles can penetrate the nucleus of the living cell and are used to deliver chemotherapeutic drugs or remove cells in cancer [78].

Nanotweezers are surgical tools, which can be used to grab single biological molecules inside the cells that can be transferred by nanotweezers established by anchoring carbon nanotubes to the electrodes [79]. Femtosecond lasers used in neurosurgery are efficient to create strength in the range of 1013/cm^2^ mostly used to correct vision in humans. The use of nanomaterial coated i.e., carbon nanotubes, and catheters minimize invasive surgery and reduce their thrombogenic effect [80]. Nanocoated (silica, polymers, or cement) implant surfaces stimulate the rapid formation of new bone. Stress shielding and bone resorption from metallic implants such as stainless steel, and cobalt chrome alloys, were significantly reduced by the use of nanophase materials. Nanolevel coating greatly improved the biocompatibility, wear characteristics, and fixation of surgical implants [81]. Hydroxyapatite nanoparticles are applied as bone cement in bone replacement surgery. Ostim^®^ (Osartis GmbH, Elsenfeld, Germany), and NanOss™ (Angstrom Medica, Woburn, MA, USA), are 100% synthetic hydroxyapatite nanoparticles with a strength of stainless steel and are fully absorbed after a few months [82,83]. VITOSSO^®^ (Orthovita-Inc, Malvern, PA, USA)contains β-tricalcium phosphate nanoparticles that resemble human cancellous bone in porosity and structure facilitating faster and increased biosorption and vascular invasion [84]. No research article is available on nanocoated implants used in cancer surgery.

## 3. Application of Nanoparticles in Modern Cancer Therapies

### 3.1. Nanoparticles in Cancer Stem Cell (CSC) Therapy

The traditional way of tumor cell removal includes cellular damage, apoptosis, or necrosis by the use of chemotherapy or radiotherapy. They are targeted toward tumor cell removal but not towards the removal of cancer stem cells (CSC) [85]. The elementary source of cancer is Cancer Stem Cells (CSCs) which are a collection of dividing cells with a high power resistance to the drugs, residency in hypoxic tumor regions, often accountable for tumor development and re-occurrence. The use of nanotechnology in CSC-based therapies is an emerging field of biomedical sciences [75]. The use of nanomedicine for the treatment of CSCs resolves stability and solubility problems and increases their cellular uptake, prolongs systemic circulation, and improved biodistribution [86]. Abolishment of CSC will provide reduced metastasis and long-lasting treatment for remission. Several nanoparticles are designed to act as an antitumor drug [87]. Albumin nanoparticles entrapped in an all-trans-retinoic acid surface coated with hyaluronic acid attack the CD44 receptor present on CSC. Silica-based nanoparticles are used as nucleus-targeted drug delivery (NTDD) to reverse CSCs drug resistance and causality of apoptosis [15,85]. Pluronic F127 micelles expand the performance and improve the CSC effectiveness of citral in breast cancer [88]. In F9 teratocarcinoma stem cells, a model used for the evaluation of cytotoxicity- and differentiation-mediated cancer therapy shows the double role of silver nanoparticles [89].

### 3.2. Nanoparticles in Immunotherapy of Cancer

Immunotherapy has developed a potent clinical approach to cancer treatment. Autoimmunity and nonspecific inflammation are the commonly observed side effects of this therapeutics. Along with that the low insufficient immunogenicity of the tumor cells are the main hurdle in obtaining promising results through immunotherapy. The adaptability and tunability of nanoparticles mark them as an auspicious podium for attending to distinct challenges faced by numerous cancers [90]. Nanoparticles are engineered to transport chemotherapeutic, tumor antigens, phototherapeutic, or whole tumor cells in a fashion to efficiently and securely activate the host’s immune organization in contradiction to tumor cells. Nanovaccines suggested a distinctive stage for the codelivery of modified tumor neoantigens and adjuvants for robust immune responses contrary to destructive tumors. Nanoparticles are engineered to elicit immunogenicity by inducing ferroptosis of the tumor cells. Ferroptosis is induced by ultrasmall iron oxide nanoparticles through the Beclin1/ATG5-dependent autophagy pathway [42]. The mixture of nanoparticle-induced ferroptosis and obstruction of programmed cell death proficiently obstruct the development of B16-F10 melanoma tumors and lung metastasis of 4T1 breast tumors, signifying the capability of ferroptosis initiation for endorsing cancer immunotherapy [91]. Immunomodulation of macrophages arose as a promising corrective approach against cancer. The attachment of signal regulatory protein alpha (SIRPα) on macrophages to CD47, a “don’t eat me” signal on cancer cells and colony-stimulating factors, secreted by cancer cells, polarize tumor-associated macrophages (TAMs) to a tumorigenic M2 phenotype. Genetically engineered cell-membrane-coated magnetic nanoparticles (gCM-MNs) can disable both mechanisms [92]. Toll-like receptor (TLR-4) based Bacillus Calmette-Guérin (BCG) and monophosphoryl lipid A (MPLA) are active immunotherapeutics that have gained FDA agreement for their clinical use in cancer treatment [41].

### 3.3. Targeted Delivery of Therapeutics Using DNA and RNA in Tumor Cells

Nanoparticles used to deliver anticancer drugs grasp considerable capacity in cancer therapy, lacking specificity. Active targeting, by use of specific ligands to functionalize nanoparticles, is drawing ample consideration in recent years. The surface area of the nanoparticles can be coated with DNA, RNA, peptides, aptamers, or antibodies to use as targeted nanotherapeutics with drug delivery. The therapeutic and diagnostic properties of nanoparticles are commonly known as theranostic. DNA nanotherapeutics involve the delivery of plasmid DNA or specific DNA strands to treat malignant tumors. Guerrero-Cázares et al. [93] created biodegradable nanoparticles of poly β-amino esters (PBAEs) and loaded them with plasmid DNA. The transfection was done in Glioblastoma multiforme mouse models. The in vitro and in vivo experiments demonstrated safe and high transfection efficiency indicating nanoparticles as effective nano vehicles for the deployment of genetic medicines. In another study, DNAzymes nanoflowers were developed for doxorubicin (Dox) delivery. These DNAzymes catalytically cleave P-glycoprotein (P-gp) mRNA which aided in the release of chemo drugs for reversing multiple drug resistance in cancer therapy [29].

RNA molecules particularly siRNA, mRNA, and microRNA (miRNA) have enormous potential for nanotherapeutics and immunomodulation of cancer. They initiate a series of adaptive and innate responses of the immune system by silencing and upregulating immune-specific genes. Although RNA therapeutics is effective in gene knockouts or regulating the expression of proteins in the therapy of the targeted cancer cells, the RNAs instability, and many physiological barriers obstruct the effective transfection and delivery. Thus, it is a clinically important task to effectively deliver all RNAs to cancer tissues or cells. To make the transfection method more reliable, nanoparticle-based delivery of RNA molecules is used in many in vivo and in vitro [94,95,96].

The development of mRNA-based therapeutic vaccines is not only capable of delivering genetic information but also induces immunostimulatory activity [97]. Restoring the function of tumor-suppressing proteins is captivating therapy for cancer treatment. A study performed by Xiao et al. [98], showed that combinatorial therapy with the help of a CXCR4-targeted p53 mRNA nanoparticle platform and anti-PD-1 therapy induced the reprogramming of cellular and molecular components of immunosuppressive tumor microenvironment in p53 deficient hepatocellular carcinoma models. Another study demonstrated the restoration of functional tumor suppressor gene, phosphatase, and TENsin homolog deleted on chromosome 10 (PTEN) through the reintroduction of PTEN mRNA encapsulated in polymer–lipid hybrid nanoparticles coated with a polyethylene glycol shell, into the PTEN deficient prostate cancer cells. A significant reduction in tumor growth was observed in vivo in mouse models with prostate cancer [99].

In the progression of cancer, dysregulation of miRNA expression occurs. The restoration of proper expression of tumor suppressor miRNAs or inhibiting overexpressed oncogenic miRNAs is a suitable strategy for cancer therapy [100]. The first nanocarrier-formulated miRNA-based drug MRX34 encapsulated in a miR-34a mimic was terminated in phase Ist clinical trial due to toxicity induced by payload miR-34a. miR-34a affected the adjacent tissues causing toxicity and death. Another miR-16 mimic (Targomi R) showed encouraging results in malignant pleural mesothelioma or non-small cell lung cancer patients (NCT02369198 trial), in which 73% of patients have attained disease control. It was encapsulated in an EnGeneIC delivery vehicle (EDV), that consists of 400 nm nonliving bacterial minicells, coated with anti-EFGR bispecific antibody, a cancer cell-targeting moiety [101].

Specifically designed siRNA is used in targeting cancer-related genes, invasion, angiogenesis, and metastasis [102]. siRNA interacts and induces the silencing of targeted genes (mRNA) post-transcriptionally. Nine anticancer nanotherapeutics that involve siRNA-encapsulated nanocarriers reached clinical trials. However only, four siRNA nanotherapeutics reached the phase II trial; DCR-MYC was terminated by the sponsor, Atu027 and TKM-PLK1 have completed phase II, while siG12D LODER is currently recruiting. No nanotherapeutics have reached phase III trials. Many siRNA nanotherapeutics showed hematologic and electrolyte abnormalities. Thus, future nanotherapeutics must examine the optimization of siRNA formulation as well as the size of nanoparticle diameter for effective cellular uptake. Co-delivery of two or more siRNA will be more effective in antitumor efficacy. Standard nanoparticle vehicles replaced with biodegradable nanoparticles will ensure safety [103,104].

### 3.4. Cell Membrane Mediated Biomimetic Nanoparticles in Cancer Therapy

Traditional treatment methods for cancer lack precise drug delivery, induce toxicity, and non-specific targetability [105]. A biomimetics is a novel approach that employs biomaterials that mimic natural biological molecules to overcome immune barriers [106]. Biomimetic nanomaterials consist of extracellular vesicles, such as exosomes, microvesicles, and cell membrane-derived vesicles such as red blood cell membrane, white blood cell membrane, platelet membrane, the cancer cell membrane can provide efficient biocompatibility and biodegradability [107]. Combinatorial therapy using polymeric drugs and cell membranes improves drug delivery efficiency without causing high toxicity [108]. Zhang et al. [62] integrated poly (lactic-co-glycolic acid) (PLGA) with the red-blood-cell membrane (RBCm) and analyzed can GA-loaded RBCm nanoparticles to maintain and upgrade the GA-induced anti-tumor ability with reduced toxic effects in the treatment of colorectal cancer comparison to free GA. It was established through in vitro results that the biomimetic nanosystem provided biocompatibility and stability in comparison with bare nanoerythrosomes. Guo et al. [109] synthesized a biomimetic nanoplatform that utilizes dendritic large pore mesoporous silica nanoparticles (DLMSNs) for efficient delivery of oxygen as well as nano radiosensitizer. The external surface was coated with citric acid for the attachment of Cu-Se-Au alloy nanoparticles that have great photothermal conversion and radiosensitizing performances. The internal surface was coated with perfluorohexane for the delivery of oxygen. White blood cells were enclosed on the DLMNs surface to target tumors and lower abrasion to normal cells. The resultant nanosystem was effective against breast cancer. Lu et al. [110] have developed CRPC (Castration-resistant prostate cancer) cell membranes as biomimetic vectors for the coating of PEG−PLGA polymer consisting of the therapeutic drug docetaxel (DTX). In vivo studies confirmed target delivery in CRPC tumors in mice models. The therapeutic efficacy was also found to be improved. Jing et al. [111] demonstrated the potential use of extracellular vesicles as nanocarriers. They synthesized a nanoprobe system, 68 Ga-L-NETA-DBCO injected extracellular vesicles, obtained from adipose-derived stem cells. PET/CT and NIRF imaging results showed uptake of tracer in the orthotopic colon cancer model suggesting its potential role in image-guided surgery.

Exosomes and microvesicles are made up of bilayers that are released from the cell membranes for communication intercellularly. Therefore, nanoparticles based on both extracellular vesicles can carry therapeutic RNAs and as well as perforate target cells to improve RNA-based cancer treatments [112]. In a study performed by Milán Rois et al. [113], gold nanoparticles are used as carriers to ensure the effective restoration of miRNA expression that was dysregulated in uveal melanoma. Yuan et al. [114] designed cancer cell membrane-camouflaged gelatin nanoparticles (CSG@B16F10) for delivery of CD73siRNA and oxygen-generating agent catalase (CAT) together thus, enhancing CD73-ADO pathway-mediated T cell immunosuppression and alleviation of tumor oxygenation. In vivo studies with mice showed PD-L1 checkpoint blockade and thus tumor suppression by ∼83%. In another study, extracellular vesicles derived from hepatocellular carcinoma were utilized as a surface nanocarrier for consecutive nanocatalysts GOD-ESIONs@EVs (GE@EVs). Glucose consumption was catalyzed by glucose oxidase (GOD) and the generation of toxic free radicals catalyzed by ESIONs under an acidic tumor microenvironment. This causes the killing of HCC cells leading to tumor suppression [115].

### 3.5. Tumor Microenvironment (TME) Targeted by Nanotherapy

Numerous cell types that constitute the TME are fibroblasts, myofibroblasts, endothelial cells, stem cells, pericytes, immune cells, and inflammatory cells. Deficiency in the concentration of oxygen, resistance to drugs, low immunogenic antigens, acidic physiological conditions, and alteration in glucose metabolism are general features observed in many cancers causing obstructions in the efficacy of chemotherapeutic treatments [116]. Nanoparticle approaches are helpful in the delivery of drugs to the cellular as well as noncellular tumor microenvironment (physiological conditions).

#### 3.5.1. Targeting Cellular Tumor Microenvironment

Normal cells contain fibroblasts that act as matrix support for normal epithelial cells. However, cancerous cells have prominent modified myofibroblasts (CAF; Cancer-associated fibroblasts) generally not found in normal cells. CAF in a stromal environment forms tumor-producing effects and prevents nanomedicine delivery by releasing pro-tumorigenic cytokines, enhancing solid tumor pressure and interstitial fluid pressure (IFP), and unspecific internalization [117]. A membrane-bound protease; Fibroblast Activation Protein (FAP), a cytoskeletal protein; α-SMA, ED-FN; a splice variant of fibronectin and vimentin are common overexpressed biological markers for inactivated CAFs [118].

A novel nanoparticle, albumin nanoparticle of paclitaxel (HSA-PTX) wrapped into the CAP-modified thermosensitive liposomes (CAP-TSL) with the incorporation of IR-780, a photothermal agent, into CAP-TSL, synthesized by Yu et al. [119]. The designed nanoparticle effectively increased the drug retention in tumors, simultaneously effluxion of HSA-PTX through FAP-α. Irradiation killed the tumor cells and fostered the penetration of HSA-PTX in its subterranean regions. The nanoparticle successfully showed an anti-tumor effect in Pan 02 subcutaneous and tumor mice. Another study done by Zhang et al. [120] demonstrated that the synergistic effect of gemcitabine-cisplatin nanoparticles on α -SMA-positive tumor-associated fibroblasts was synergically effective against tumor of bladder carcinoma. Wang et al. [121] developed a dual-target drug delivery system with the combination of paclitaxel (PTX)-loaded poly(ethylene glycol)-poly(lactic acid) nanoparticles (NPS) and a cyclic peptide (CNPs-PTX) inclining with platelet-derived growth factor/platelet-derived growth factor receptor (PDGFR-β) overexpressed on both cancer-associated fibroblasts and myeloma cells. The cytotoxic analysis demonstrated toxicity in both CAF and myeloma cells led by cyclic peptide CNPs-PTX was more vigorous than PTX-loaded conventional NPs (NPs-PTX). Another study investigated the potential of P selection molecules involved in metastasis by localizing nanomedicines at tumor-affected endothelial cells. A fucosylated polysaccharide with nanomolar affinity to P-selectin; a drug delivery platform was created. The designed nanoparticle effectively localized in the tumor environment and targeted MEK (mitogen-activated protein kinase) inhibitor in tumor regions, causing antitumor effects [122]. Research led by Kuo et al. [123] demonstrated the downregulation of inhibitors of apoptosis proteins (XIAP and cIAP) and upregulation of caspase-3 expression by BV6- and GDC0152-encapsulated solid lipid nanoparticles (SLNs) with surface transferrin (Tf) and folic acid (FA) (BV6-GDC0152-Tf-FA-SLNs) in brain cancer stem cells of human.

#### 3.5.2. Targeting Non-Cellular Tumor Microenvironment and Physiological Conditions

Hypoxia occurs due to the inability to meet the need for oxygen in cancer cells due to immature vasculature. Hypoxic regions in solid tumors are difficult to treat because of limited drug circulation and the absence of oxygen-free radicals in chemo- and radiation therapy, respectively [124]. In recent research, a hybrid sonosensitizer developed from a photosynthetic microorganism, cyanobacteria (Cyan) attached with ultrasmall oxygen-deficient bimetallic oxide Mn_1.4_WOx nanosonosensitizers (M@C), was developed to overcome resistance induced by hypoxic conditions in tumor [125]. With the assistance of nanosonosensitizers, the production of ROS was increased against cancer cells under ultrasound irradiation. Another MnO_2_-based nanoparticle (HMIB NPs) was constructed to attain great phototherapeutic efficiency, by NIR light mediation, oxygen self-supply, and deep diffusion through tumor microenvironment response. The immunofluorescence analysis showed that HMIB NPs not only provide oxygen in the tumor microenvironment for lowering hypoxia but also decreased tumor microenvironment responsive size for the improvement of deep intratumoral diffusion [126].

The acidic microenvironment in cancer cells that arise due to increased glycolytic rate, has emerged as a multifaceted target for theranostic platforms [24]. Zhang et al. [127], developed a pH-sensitive system based on propylene glycol alginate sodium sulfate (PSS) that has anti-platelet aggregation ability. PSS@DC nanoparticles were designed in which chemotherapeutic drug doxorubicin (DOX) and celecoxib (CXB) compiled to form nanocores, hydrophobic in nature and PPS encapsulated these nanocores in conjunction with DOX via a benzoic-imine linker and treated in mice breast cancer model. The nanoparticles demonstrated distinct pH sensitivity and increased the efflux of DOX at the acidic pH mimicking the tumor microenvironment. Another study done by Chu et al. [128], showed biodegradable iron-doped ZIF-8 nanocrystals that can be degraded under an acidic tumor microenvironment, leading to the release of DOX (doxorubicin) and photothermal transforming agent, ICG (indocyanine green) at high speed. DOX and ICG demonstrated chemotherapeutic efficacy and NIR-triggered photothermal therapy. A fatality rate was 93.7% observed for cancer cells, which provides promising results in cancer treatments.

Alterations in metabolic pathways are one of the remarkable features of cancer. Liu et al. [129] developed an amorphous iron oxide nanoparticle (NP)-based RNAi system that modulated the glycolysis pathway by suppressing MCT4 expression, induced acidosis in tumor cells, enhanced oxidative stress in tumor cells via the Fenton-like reaction, reducing tumor growth. Another research led by Yu et al. [130] showed inhibition of aerobic glycolysis by providing enough oxygen to facilitate anti-metastasis with the assistance of enzyme-powered nanomotor (NM-si).

## 4. Toxicity and Future Challenges of Nanoparticles in Cancer Therapy

The properties of small size, individual tensility, high reactivity, and magnetic properties of nanomaterials have elevated anxieties about consequences on health, environment, and safety. However, the widely held available data indicate that there is nothing exclusively toxic about nanoparticles as a class of materials. The low biocompatibility of the nanomaterials used to design nanoparticles most commonly shows some toxic effects. Recently there has been a certain unresolved debate on the possible toxicity of a specific type of nanomaterials such as carbon nanotubes, nanogel, and chemo nanotherapeutics with tissue injury in animal studies. The use of carbon nanotubes shows higher toxic potentials and was found to be carcinogenic for the lung, gastrointestinal tract, central nervous system, and blood [131]. Silver-based NPs toxicity has been reported at the cellular level in the terms of ROS generation, DNA damage, and cytokine induction in vitro conditions. Only a few in vivo studies on silver nanoparticles have been reported showing possibilities of, antagonistic effects on hepatic, circulatory, central nervous, dermal, and the respiratory system [132]. Studies on the physicochemical properties of Ag NPs will help to assess their effects on cellular uptake and bioavailability. Some heavy metals used in nanoparticles can be stored in the kidney and liver accelerating the toxicity [133]. Silicate-based nanoparticles are categorized by a protruding growth in the liver and lung causing fibrosis, an important side effect [134]. A bibliometric analysis was used by Zhong et al. [10] to evaluate the injuriousness of quantum dots. The toxicity has been studied in male mouse reproductive and sexual behaviors and on the health of offspring. Its finding indicates the low toxicity of graphene quantum dots in germ cells and their fast elimination through urine and/or feces. High doses of it are almost imperceptible to the brain, testis, and epididymis of male mice [135]. The side effects on lipoproteins, serum proteins, and the extracellular matrix of the liver and kidney have been also reported from the use of more biocompatible and mostly clinically used nanoparticle liposomes.

Despite numerous aptitudes of nanoparticles in different therapeutics approaches to cancer, only limited studies have inspected the adverse human body’s response to nanoparticle contact. Similarly, few studies have discovered the probable responses towards the uninhibited application of nanoparticles on the health of surrounding tissues of tumors. Hence, there is an evident requisite to promote research in this area. There is a serious demand to discover the properties and machinery of these elements in humans, such as inflammation, DNA interaction, and adverse findings on diverse organs, tissues, and cells. Intensive care is needed to find out the environmental effects of it and adequate techniques are required to calculate the exposure status of nanoparticles. Gene expression profiling of tumor-surrounded cells can deliver evidence on the probable act of nanoparticles and their human significance. The task will be very complex but informative due to the diversity of groups of molecules present in nanoparticles, unlike assets and effects.

## 5. Conclusions

Nanotherapeutics have now generated a significant revolution in the treatment of cancer and seem to have the probability to produce many others. This review has explored the importance of the convergence of nanotechnology with a different commonly used method for cancer therapy for a more successful outcome of treatment. To deal with the safety concern for humans, and the environment, a wide and determined direction of research based on nanotechnology must be carefully understood and its risks should be assessed. The knowledge of researchers in different fields is required to be conveyed to each other to move new findings out of the laboratory into clinical practices where patients can acquire profits. An understanding of the bio-interaction of nanoparticles, their universal transportation towards tumor cells and their direction towards the microenvironment will help to develop safer and more efficacious nanotherapeutics. More research is required to develop the next generation of nanotherapeutics with the incorporation of new molecular entities, such as kinase inhibitors, siRNA, mRNA, and gene editing. We assume that nanotherapeutics will dramatically improve patient survival, move the model of cancer treatment and develop certainty in the predictable future.

## Figures and Tables

**Figure 1 cancers-15-00162-f001:**
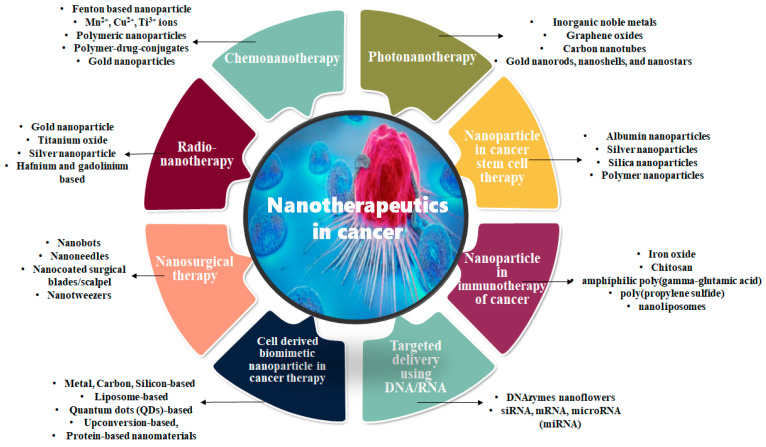
Nanoparticles in different types of cancer therapies.

**Table 1 cancers-15-00162-t001:** A different class of nanoparticles, their characteristics, mechanism of function, and application in cancer management.

S.No	Nanoparticles	Characteristics	Mechanisms	Applications in Cancer Therapy	References
**A.**	**Inorganic**				
1.	Zinc oxide nanoparticles	Electrical resistivity, Optical characteristics, Dynamic capabilities	Penetration of cell wall, Generation of reactive oxygen species	Chemotherapy	[6]
2.	Copper and cuprous oxide nanoparticles	Antimicrobial properties, High temperature photo-catalytic properties	Penetrate cell membrane, Destroy exposed cells	Chemotherapy	[7]
3.	CaO nanoparticles	Structural and optical properties, Adsorbent	Inhibit the biofilm formation Create an artificial calcium overloading, stress in tumor cells, Cell death	Chemotherapy	[8]
4.	CeO_2_ nanoparticles	Photocatalytic degradation of pollutants, n-type semiconductor, Redox property, Strong absorption of light, Stability, Nontoxicity	Attachment and penetration in the cell wall, Cause inhibition of RNA and DNA	Radiotherapy	[9]
5.	Al_2_O_3_ nanoparticles	High thermal and low electrical conductivity, Highly flammable, An irritant	Facilitate conjugative transfer of antibiotic resistance genes, Target autophagy signaling in cancerous cells	Chemotherapy	[10]
6.	Carbon-based nanoparticles (graphene)	Heat and electrical conductivity, Mechanical properties	Activate components of the human immune system and cause immunogenicity	Chemotherapy	[11]
7.	Quantum dots	Semiconductor, Stability at a higher temperature, High brightness, Broad spectrum of absorption, Resistant to chemical degradation, Narrow emission bands, High photostability	Early diagnosis of cancer, In vitro and In vivo tumor imaging, Targeted gene delivery, Used in photodynamic therapy, Unique carrier for drug delivery breast	Photothermal therapy	[12]
8.	TiO_2_ nanoparticles	Bio-compatibility Low cost and high stability, Enhance permeability and retention effect	Partial decomposition of the membrane wall, Enter in the cell Production of ROS, Peroxidation processes, Cell dyeing	Phototherapy	[13,14]
9.	Ag and Ag_2_O nanoparticles	p-type semiconductors Photo-catalytic Photochemical Biological synthesis	Release of ions that destroy cell membrane, Cell death	Photothermal therapy	[15]
10.	Silica-based nanoparticles	Biocompatibility, High surface area	Induce ROS, Autophagy dysfunction	Photochemotherapy	[16]
11.	Super magnetic iron oxide nanoparticles	High magnetization, Targeted release of drugs	Detect receptors on the surface of cancer cells Detect unusual angiogenesis in the tumor microenvironment, Detect circulating tumor cells Detect soluble tumor biomarkers	Photothermal Therapy	[17]
12.	Nanoshells	Dielectric silica based, Upconversion from light to heat energy Electrostatic stabilization	Destroy tumor cells, Heat energy destroys cancer cell	Immunotherapy	[18]
13.	Gold-based nanoparticles	Biocompatibility Optoelectronic properties, Peak optical density	Generation of ROS, Release of cytokines, Apoptosis	Chemo-photo thermal therapy	[19]
14.	Calcium Phosphate Nanoparticles	Biologically compatible, Biodegradable	Decrease toxicity and enhance transfection features	Immunotherapy	[20]
**B.**	**Organic**				
1.	Polymeric nanoparticles	Drug delivery, Non-biodegradable polymers, bioimaging, Anti-inflammatory activity, Anti-glioma activity	Facilitates endocytosis, Ligand-based targeting mechanism	Photodynamic therapy	[21]
2.	Nanostructured lipid carriers	Enhance solubility, Improve storage stability, Increase permeability and physiological bioavailability, Low side effects, Longer shelf-life	Target tissue delivery, Active or passive targeting	Photodynamic therapy	[22]
3.	Solid lipid nanoparticles	Colloidal nanocarriers, Micelle-like structure, Assist in the association of the ionic components with plasma and endosomal lipid membrane system	Reduce the development of doxorubicin-sensitive breast cancer cells (MCF-7)	Immunotherapy, Chemotherapy	[23,24]
4.	Self-assembled nanomaterials	Selective tumor accumulation, Enhance specificity in nanocarrier communication with the target cell	Chemotherapeutic efficiency increase, Induce apoptosis in in vitro and in vivo, Reduce toxicity to nearby cells	Phototherapy, chemotherapy, immunotherapy	[11]
5.	Nanogels	Charge Size (10–100 nm), porosity, Amphiphilicity and degradability, Softness	Colloidal stability increases with the interaction of inorganic nanosystems, Upgrade aqueous solubility, Protection against the mononuclear phagocytic system (MPS). Target the site of interest by conjugating targeting ligands on its surface	Immunotherapy	[25]
6.	Nanoemulsion	Emulsifying agents as well as oil, Optical clarity Stability, Biodegradability	Enhance site specificity, Enhance the therapeutic effectiveness of the drug, Lowers toxic effects on adjacent cell multidrug	Chemotherapy	[26]
7.	Nanocapsules	encapsulate drugs with specific chemical receptors	Bind to specific target cells. Receptor-specificity to target cancer breast	Geno therapy, Chemotherapy and Photothermal therapy	[27]
8.	Dendrimers	Highly branched, Easily modifiable surfaces	Association of drugs with nucleic acids (DNA or RNA)	Photodynamic therapy, Radio-nanotherapy	[28,29]
9.	DNA nano-cocoons	Self-assembled single-stranded DNA	Binding of the specific receptors on the surface of a cancer cell Acidic environment within the cancer cells causes the breakdown of the polymeric coat and unleashing a massive dose of the drug load	Immunotherapy	[30]
10.	Chitosan nanoparticles	Biodegradability and Biocompatibility Nontoxicity	Destruction of cell membrane and release of drug	Photodynamic, Photothermal, Chemotherapy	[31,32]
11.	Triclosan	Antimicrobial properties. Anticancerous properties chlorinated aromatic compounds Contains functional group both functional group and phenols	Inhibit a specific target. Ability to inhibit fatty acid synthesis. Induce apoptosis in this prostate cancer oral	Immunotherapy	[33]
12.	Peptides	Small chains of amino acids attached by peptide bond linkage Cationic and amphipathic peptides	Destruction of the lipid bilayer structure. Attach to the DNA and RNA and constraint replication	Immunotherapy	[34]
13.	Liposomes	Biocompatible Biodegradable Stable in colloidal solutions Higher anti-tumor efficacy Enhanced bioavailability Cytotoxic drugs delivered in amounts at the tumor site	Increase in temperature of the tumor, cause agglomeration of liposomes	Photodynamic, Radiotherapy	[35,36]
14.	mAb nanoparticles	Specific targeting ability, Antitumor effect. Lower toxicity	Increase chemotherapeutic effects of anticancer drugs Conjugated mAbs with cytotoxic drugs Ex-Trastuzumab (Herceptin)	Immunotherapy, Chemotherapy	[37,38]
15.	Exosomes	Biocompatibility	Gene therapy can induce cell death by deliver ing transgene or cell death-triggering gene to tumor cells	Immunotherapy	[39]
16.	Micro-vesicle based particles.	transparency, absorption, luminescence and scattering	Gene therapy can induce cell death by deliver ing transgene or cell death-triggering gene to tumor cells	Phototherapy, Chemotherapy	[40]
17.	Cyclodextrin Nanosponges	crystalline or amorphous structure and spherical shape or swelling properties	Cyclodextrin-based nanosponges can form complexes with different types of lipophilic or hydrophilic molecules	Chemotherapy	[41]
**C.**	**Mixed**				
1.	Nanoscale Cordination Polymer (NCPs) or Nanoscale metal-organic frameworks (NMOFs)	metal ions or clusters with linking ligands compositional and structural tenability highly porous and oriented structures intrinsic biodegradability	nanocarriers for anticancer drug delivery efficient loading of diverse cargos	Immunotherapy	[42,43]
2.	polysilsesquioxane (PSQ)	formed via hydrolysis and condensation bis(trialkoxysilanes) ((R’O)_3_-Si-R-Si-(OR’)_3_) via sol-gel reactions	allow much higher drug loadings than silica-based materials	Chemotherapy, Radiotherapy	[44]
3.	Combination of biodegradable polymers with silica, gold, or iron oxide nanoparticles. Example-Hybrid mesoporous silica (MSN), gold, and iron oxide nanoparticles	Having hundreds of empty channels or mesopores peptide target ligands or small interfering RNAs can be biodegradable polymers	encapsulate and/or absorb bioactive molecules useful for drug delivery	Photodynamic therapy	[45]

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
