# Peer review of "Nanoparticles and Nanomaterials-Based Recent Approaches in Upgraded Targeting and Management of Cancer: A Review"

_cancers, 2022, doi:10.3390/cancers15010162_

Round 1
Reviewer 1 Report
cancers-2032230
Entitled: Nanoparticles in upgraded targeting and management of Cancer
The submitted review deals with many different nano-formulations and various nanotherapeutic applications, as mentioned in the introduction. Nevertheless, the scope, respectively the structure, of this review is a bit diffuse. It should by highlighted that applications and therapeutic concepts are introduced and the review is not focused on composition and production related aspects.
In detail I would recommend to position figure 1 before table 1 thus the readers are aware of the following chapters and the general scope.
Table 1 should be reformatted by removing the bullet points, controlling the style, and typing errors. Not all types of nanoparticles, named in the following text are listed in the table, such as exosomes and micro-vesicle based particles. Furthermore, many publications are dealing with different types of nanoparticles therefore it´s not clear why only few of them are referred.
In chapter 2, entitled Application of nanoparticles in frequently applied cancer therapies, the authors describe different treatment principles combined with nanoparticles thus the title should be changed to reflect the content. Some examples are more detailed described as others and also phrasing need to be revised.
For chapter 3, entitled Application of nanoparticles in modern cancer therapies, I would also recommend to change the title such as replacing modern by innovative.
In chapter 4, toxicity and selected future challenges are addressed. In this respect also the toxicity and challenges of organic nanoparticles should be considered. The statement -Despite numerous aptitudes of nanoparticles in different therapeutics approaches to cancer, only limited studies have inspected the adverse human body’s response to nanoparticle contact- is only partially true and related publications should be included. Furthermore, especially in the treatment of cancer the risk-benefit ratio of nanotherapeutics should be outlined.
Only in some examples the application performance is described, such as in vitro and in vivo, cell assays and animal models. It should be comprehensively completed to distinguish between research and clinical results. In this respect a table of regulatory approved nanotherapeutics would be very informative.
In conclusion, the submitted manuscript provide a profound overview. However, the manuscript should be thoroughly revised according phrasing and typing errors as well as recommendation mentioned above.
Author Response
Response to reviewer’s -1 comment
Table 1 should be reformatted by removing the bullet points, controlling the style, and typing errors. Not all types of nanoparticles, named in the following text are listed in the table, such as exosomes and micro-vesicle based particles. Furthermore, many publications are dealing with different types of nanoparticles therefore it´s not clear why only few of them are referred.
We are thankful to the reviewers for their valuable comments on the present manuscript. The table is reformatted by removing the bullet points, controlling the style and typing errors. Exosomes, vesicle-based particles, and cyclodextrin nanosponges have been added to the revised manuscript. The vastness of the content hinders us to include all the publications related to the nanoparticles. Therefore we have tried to include the recent relevant references. However, some new references have been added to the appropriate place.
In chapter 2, entitled Application of nanoparticles in frequently applied cancer therapies, the authors describe different treatment principles combined with nanoparticles thus the title should be changed to reflect the content. Some examples are more detailed described as others and also phrasing need to be revised.
The suggested comment has been incorporated in the revised manuscript. The title changed to “Principles of application of nanoparticles in frequently applied cancer therapies.” Some topics of chapter 2 have been studied by various researchers and published valuable data on the mode of action and activity of nanoparticles in cancer therapy. We have tried to incorporate most of the work done by them. So they become vast while only a few studies have been done on some topic so they become short. The phrasing in the manuscript has been revised in the revised manuscript.
For chapter 3, entitled Application of nanoparticles in modern cancer therapies, I would also recommend to change the title such as replacing modern by innovative.
In the revised manuscript we have replaced the term modern with innovative.
In chapter 4, toxicity and selected future challenges are addressed. In this respect also the toxicity and challenges of organic nanoparticles should be considered. The statement - Despite numerous aptitudes of nanoparticles in different therapeutics approaches to cancer, only limited studies have inspected the adverse human body’s response to nanoparticle contact- is only partially true and related publications should be included. Furthermore, especially in the treatment of cancer the risk-benefit ratio of nanotherapeutics should be outlined.
In the present manuscript, we have incorporated a few studies done on the toxicity assessment of nanoparticles (Sharma et al 2021; Koyande et al 2022). I totally agree with the reviewer's concern over reporting on the risk-benefit ratio of nanotherapeutics. We were unable to find any relevant research article on that. If you can suggest something on that we will be thankful to you.
Only in some examples the application performance is described, such as in vitro and in vivo, cell assays and animal models. It should be comprehensively completed to distinguish between research and clinical results. In this respect a table of regulatory approved nanotherapeutics would be very informative.
Respected reviewer, I totally agree with your suggestion on the incorporation of the table on regulatory-approved nanotherapeutics. We are planning to make a full review article on the in vitro, in vivo, and clinical outcome of various nanotherapeutics and their status in regulatory approval.
In conclusion, the submitted manuscript provide a profound overview. However, the manuscript should be thoroughly revised according phrasing and typing errors as well as recommendation mentioned above.
In light of the reviewers' comments, the present review has been thoroughly revised accordingly. Phrasing and typing errors as well as the recommendation mentioned above were incorporated in the revised manuscript.

Reviewer 2 Report
In this review article, the authors summarized applications of various nanoparticles to cancer treatments. Although there are many review articles of the same topic, this contains the description of recent advances. Thus, it is worth to be published. However, I think that the topics of this manuscript, application of inorganic to organic nanoparticles to a variety of therapy, is too broad. I recommend that the authors should focus the application or the nanoparticle and improve the title to which readers pay attentions. Besides, Table 1 should be improved: The applications and the references should be matched. (Each application should be cited.) And, the whole manuscript should be checked to improve English.
Author Response
In this review article, the authors summarized applications of various nanoparticles to cancer treatments. Although there are many review articles of the same topic, this contains the description of recent advances. Thus, it is worth to be published. However, I think that the topics of this manuscript, application of inorganic to organic nanoparticles to a variety of therapy, is too broad. I recommend that the authors should focus the application or the nanoparticle and improve the title to which readers pay attentions. Besides, Table 1 should be improved: * P 0 WORDS File Edit View Insert Format Tools Table Help Paragraph The applications and the references should be matched. (Each application should be cited.) And, the whole manuscript should be checked to improve English.
In light of the reviewer's comment, the title of the manuscript changed to “Nanoparticles application in upgraded targeting and management of Cancer: A Review” which emphasizes the application of various nanoparticles in cancer management.
In the revised manuscript Table 1 has been improved and some new types of nanoparticles were added such as inorganic calcium phosphate nanoparticles, organic exosomes, micro-vesicle based particles, and cyclodextrin nanosponges. Some new references were also added related to the application of various nanoparticles.
We have checked the application and the references match and were corrected accordingly.
The revised manuscript has been checked by the Grammarly software and was found 84%.

Round 2
Reviewer 2 Report
I pointed out that the topic of this review is too broad. However, the authors did not focus on any topics. Instead, they added some other nanoparticles to the revised version. If the authors would like to introduce recent various nanoparticles in biomedical applications, recent advances in this field should be clarified.
The authors described that nanoparticles are useful for various applications such as chemotherapy, phototherapy, immunotherapy radiotherapy... and so on, as shown in Figure 1. The authors should show for which application each nanoparticle shown in Table 1 is used. Types of cancer are listed in Table 1 as "application". But, chemotherapy, phototherapy, immunotherapy radiotherapy.. and so on should be also shown in Table 1. Reference numbers should be added in the main text and Table 1 instead of the first author and the publication year.
Author Response
Response to Reviewer’s-2 comments
In this review article, the authors summarized applications of various nanoparticles to cancer treatments. Although there are many review articles of the same topic, this contains the description of recent advances. Thus, it is worth to be published. However, I think that the topics of this manuscript, application of inorganic to organic nanoparticles to a variety of therapy, is too broad. I recommend that the authors should focus the application or the nanoparticle and improve the title to which readers pay attentions. Besides, Table 1 should be improved: * P 0 WORDS File Edit View Insert Format Tools Table Help Paragraph The applications and the references should be matched. (Each application should be cited.) And, the whole manuscript should be checked to improve English.
In light of the reviewer's comment, the title of the manuscript changed to “Nanoparticles application in upgraded targeting and management of Cancer: A Review” which emphasizes the application of various nanoparticles in cancer management.
In the revised manuscript Table 1 has been improved and some new types of nanoparticles were added such as inorganic calcium phosphate nanoparticles, organic exosomes, micro-vesicle based particles, and cyclodextrin nanosponges. Some new references were also added related to the application of various nanoparticles.
We have checked the application and the references match and were corrected accordingly.
The revised manuscript has been checked by the Grammarly software and was found 84%.

Round 3
Reviewer 2 Report
The authors did not understand my comments properly. See the attached file, and correct the manuscript.

Author Response
Comments and Suggestions for Authors (Round- 2)
I pointed out that the topic of this review is too broad. However, the authors did not focus on any topics. Instead, they added some other nanoparticles to the revised version. If the authors would like to introduce recent various nanoparticles in biomedical applications, recent advances in this field should be clarified.
The authors described that nanoparticles are useful for various applications such as chemotherapy, phototherapy, immunotherapy radiotherapy... and so on, as shown in Figure 1. The authors should show for which application each nanoparticle shown in Table 1 is used. Types of cancer are listed in Table 1 as "application". But, chemotherapy, phototherapy, immunotherapy radiotherapy. and so on should be also shown in Table 1. Reference numbers should be added in the main text and Table 1 instead of the first author and the publication year.
I totally agree with the reviewers’ comments. Our main aim of present review is to compile the types of various nanoparticles, their mode of action and their use in various types of cancer therapy. We have revised the title of manuscript with new the title i.e. “Nanoparticles and nanomaterials-based recent approaches in upgraded targeting and management of Cancer: A Review” to emphasize our aim.
The table has been modified according to suggestion’s received from the reviewer. Application of each nanoparticle in different types of therapy has been mentioned in revised manuscript. Previously I could not understand the suggestions.
Comments and Suggestions for Authors (Round 3)
The authors did not understand my comments properly. See the attached file, and correct the manuscript.
In the light of reviewer 2 (round 3 comments and suggestions) the present review has been revised again. Table has been modified. Organic, inorganic and mixed nanoparticles use in various type of cancer therapy has been included along with the no. of references instead of the first author and the publication year.
